# Benefits of Additive Noise
# in Composing Classes with Bounded Capacity

**Alireza Fathollah Pour**
Department of Computing and Software
McMaster University
fathola@mcmaster.ca

**Hassan Ashtiani**
Department of Computing and Software
McMaster University
zokaeiam@mcmaster.ca

## Abstract

We observe that given two (compatible) classes of functions $\mathcal{F}$ and $\mathcal{H}$ with small capacity as measured by their uniform covering numbers, the capacity of the composition class $\mathcal{H} \circ \mathcal{F}$ can become prohibitively large or even unbounded. We then show that adding a small amount of Gaussian noise to the output of $\mathcal{F}$ before composing it with $\mathcal{H}$ can effectively control the capacity of $\mathcal{H} \circ \mathcal{F}$, offering a general recipe for modular design. To prove our results, we define new notions of uniform covering number of random functions with respect to the total variation and Wasserstein distances. We instantiate our results for the case of multi-layer sigmoid neural networks. Preliminary empirical results on MNIST dataset indicate that the amount of noise required to improve over existing uniform bounds can be numerically negligible (i.e., element-wise i.i.d. Gaussian noise with standard deviation $10^{-240}$).[1][2]

## 1 Introduction

Let $\mathcal{F}$ be a class of functions from $\mathcal{X}$ to $\mathcal{Y}$, and $\mathcal{H}$ a class of functions from $\mathcal{Y}$ to $\mathcal{Z}$. Assuming that $\mathcal{F}$ and $\mathcal{H}$ have bounded "capacity", can we bound the capacity of their composition, i.e., $\mathcal{H} \circ \mathcal{F} = \{h \circ f \mid f \in \mathcal{F}, h \in \mathcal{H}\}$? Here, by capacity we mean learning-theoretic quantities such as VC dimension, fat-shattering dimension, and (uniform) covering numbers associated with these classes (see Vapnik (1999); Anthony et al. (1999); Shalev-Shwartz and Ben-David (2014); Mohri et al. (2018) for an introduction). Being able to control the capacity of composition of function classes is useful, as it offers a modular approach to design sophisticated classes (and therefore learning algorithms) out of simpler ones. To be concrete, we want to know if the uniform covering number (as defined in the next section) of $\mathcal{H} \circ \mathcal{F}$ can be "effectively" bounded as a function of the uniform covering numbers of $\mathcal{F}$ and $\mathcal{H}$.

The answer to the above questions is true when $\mathcal{F}$ is a set of binary valued functions (i.e., $\mathcal{Y} = \{0,1\}$ in the above). More generally, the capacity of the composition class (as measured by the uniform covering number) can be bounded as long as $|\mathcal{Y}|$ is relatively small (see Proposition 7). But what if $\mathcal{Y}$ is an infinite set, such as the natural case of $\mathcal{Y} = [0,1]$? Unfortunately, in this case the capacity of $\mathcal{H} \circ \mathcal{F}$ (as measured by the covering number) can become unbounded (or excessively large) even when both $\mathcal{F}$ and $\mathcal{H}$ have bounded (or small) capacities; see Propositions 8 and 9.

Given the above observation, we ask whether there is a general and systematic way to control the capacity of the composition of bounded-capacity classes. More specifically, we are interested in the case where the domain sets are multi-dimensional real-valued vectors (e.g., $\mathcal{X} \subset \mathbb{R}^d$, $\mathcal{Y} \subset \mathbb{R}^p$, $\mathcal{Z} \subset \mathbb{R}^q$). The canonical examples of such classes are those associated with neural networks.

---

[1]For the full version of this paper see Pour and Ashtiani (2022).

[2]The source codes are available at https://github.com/fathollahpour/composition_noise

36th Conference on Neural Information Processing Systems (NeurIPS 2022).

A common approach to control the capacity of $\mathcal{H} \circ \mathcal{F}$ is assuming that $\mathcal{H}$ and $\mathcal{F}$ have bounded capacity and $\mathcal{H}$ consists of Lipschitz functions (with respect to appropriate metrics). Then the capacity of $\mathcal{H} \circ \mathcal{F}$ can be bounded as long as $\mathcal{H}$ has a small "global cover" (see Remark 13). This observation has been used to bound the capacity of neural networks in terms of the magnitude of their weights (Bartlett, 1996). More generally, the capacity of neural networks that admit Lipschitz continuity can be bounded based on their group norms and spectral norms (Neyshabur et al., 2015, Bartlett et al., 2017, Golowich et al., 2018). One benefit of this approach is that the composition of Lipschitz classes is still Lipschitz (although with a larger Lipschitz constant).

While building classes of functions from composition of Lipschitz classes is useful, it does not necessarily work as a general recipe. In fact, some commonly used classes of functions do not admit a small Lipschitz constant. Consider the class of single-layer neural networks defined over bounded input domain $[-B, B]^d$ and with the sigmoid activation function. While the sigmoid activation function itself is Lipschitz, the Lipschitz constant of the network depends on the magnitude of the weights. Indeed, we empirically observe that this can turn Lipschitzness-based bounds on the covering number of neural networks worse than classic VC-based bounds.

Another limitation of using Lipschitz classes is that they cannot be easily "mixed and matched" with other (bounded-capacity) classes. For example, suppose $\mathcal{F}$ is a class of $L$-Lipschitz functions (e.g., multi-layer sigmoid neural networks with many weights but small magnitudes). Also, assume $\mathcal{H}$ is a non-Lipschitz class with bounded uniform covering number (e.g., one layer sigmoid neural network with unbounded weights). Then although both $\mathcal{F}$ and $\mathcal{H}$ have bounded capacity, $\mathcal{H} \circ \mathcal{F}$ is not Lipschitz and its capacity cannot be generally controlled.

We take a different approach for composing classes of functions. A key observation that we make and utilize is that adding a little bit of noise while "gluing" two classes can help in controlling the capacity of their composition. In order to prove such results, we define and study uniform covering numbers of random functions with respect to total variation and Wasserstein metrics. The bounds for composition then come naturally through the use of data processing inequality for the total variation distance metric.

**Contributions and Organization.**

- Section 3 provides the necessary notations and includes the observations that composing real-valued functions can be more challenging than binary valued functions (Propositions 7, 8, and 9).

- In Section 4, we define a new notion of covering number for random functions (Definition 10) with respect to total variation (TV) and Wasserstein distances.

- The bulk of our technical results appear in Section 5. These include a composition result for random classes with respect to the TV distance (Lemma 18) that is based on the data processing inequality. We also show how one can translate TV covering numbers to conventional $\|.\|_2$ counterparts (Theorem 17) and vice versa (Corollary 21). A useful tool is Theorem 20 which exploits kernel density estimation techniques to translate Wasserstein covers to TV covers when we add Gaussian noise to the output of functions.

- Section 6 provides a stronger type of covering number for classes of single-layer noisy neural networks with the sigmoid activation function (Theorem 25).

- In Section 7, we use the tools developed in the previous sections and prove a novel bound on the $\|.\|_2$ covering number of noisy deep neural networks (Theorem 26).

- In Section 8 we define NVAC, a metric for comparing generalization bounds (Definition 28) based on the number of samples required to make the bound non-vacuous.

- We offer some preliminary experiments, comparing various generalization bounds in Section 9. We observe that even a negligible amount of Gaussian noise can improve NVAC over other approaches without affecting the accuracy of the model on train or test data.

## 2 Related work

Adding various types of noise have been empirically shown to be beneficial in training neural networks. In dropout noise (Srivastava et al., 2014) (and its variants such as DropConnect (Wan et al.,

2013) the output of some of the activation functions (or weights) are randomly set to zero. These approaches are thought to act as a regularizer. Another example is Denoising AutoEncoders (Vincent et al., 2008) which adds noise to the input of the network while training stacked autoencoders.

There has been efforts on studying the theory behind the effects of noise in neural networks. Jim et al. (1996) study the effects of different types of additive and multiplicative noise on convergence speed and generalization of recurrent neural networks (RNN) and suggest that noise can help to speed up the convergence on local minima surfaces. Lim et al. (2021) formalize the regularization effects of noise in RNNs and show that noisy RNNs are more stable and robust to input perturbations. Wang et al. (2019) and Gao and Zhou (2016) analyze the networks with dropout noise and find bounds on Rademacher complexities that are dependent on the product of norms and dropout probability. It is noteworthy that our techniques and results are quite different, and require a negligible amount of additive noise to work, while existing bounds for dropout improve over conventional bounds only if the amount of noise is substantial. Studying dropout noise with the tools developed in this paper is a direction for future research.

Studying PAC learning and its sample complexity is by now a mature field; see Vapnik (1999); Shalev-Shwartz and Ben-David (2014); Mohri et al. (2018). In the case of neural networks, standard Vapnik-Chervonenkis-based complexity bounds have been established (Baum and Haussler, 1988; Maass, 1994; Goldberg and Jerrum, 1995; Vidyasagar, 1997; Sontag et al., 1998; Koiran and Sontag, 1998; Bartlett et al., 1998; Bartlett and Maass, 2003; Bartlett et al., 2019). These offer generalization bounds that depend on the number of parameters of the neural network. There is also another line of work that aims to prove a generalization bound that mainly depends on the norms of the weights and Lipschitz continuity properties of the network rather than the number of parameters (Bartlett, 1996; Anthony et al., 1999; Zhang, 2002; Bartlett, 1996; Neyshabur et al., 2015; Bartlett et al., 2017; Neyshabur et al., 2018; Golowich et al., 2018; Arora et al., 2018; Nagarajan and Kolter, 2018; Long and Sedghi, 2020). We provide a more detailed discussion of some of these results in Appendix H. Finally, we refer the reader to Anthony et al. (1999) for an introductory discussion on this subject.

The above-mentioned bounds are usually vacuous for commonly used data sets and architectures. Dziugaite and Roy (2017) (and later Zhou et al. (2019)) show how to achieve a non-vacuous bound using the PAC Bayesian framework. These approaches as well as compression-based methods (Arora et al., 2018) are, however, examples of "two-step" methods; see Appendix H for more details. It has been argued that uniform convergence theory may not fully explain the performance of neural networks (Zhang et al., 2021; Nagarajan and Kolter, 2019). One conjecture is that implicit bias of gradient descent (Gunasekar et al., 2017; Arora et al., 2019; Ji et al., 2020; Chizat and Bach, 2020; Ji and Telgarsky, 2021) can lead to benign overfitting (Belkin et al., 2018, 2019; Bartlett et al., 2020); see Bartlett et al. (2021) for a recent overview.

In a recent line of work, generalization has been studied from the perspective of information theory (Russo and Zou, 2016; Xu and Raginsky, 2017; Russo and Zou, 2019; Steinke and Zakynthinou, 2020), showing that a learning algorithm will generalize if the (conditional) mutual information between the training sample and the learned model is small. Utilizing these results, a number of generic generalization bounds have been proved for Stochastic Gradient Langevin Descent (SGLD) (Raginsky et al., 2017; Haghifam et al., 2020) as well as Stochastic Gradient Descent (SGD) Neu et al. (2021). Somewhat related to our "noise analysis", these approaches (virtually) add noise to the parameters to control the mutual information. In contrast, we add noise between modules for composition (e.g., in between layers of a neural network). Furthermore, we prove uniform (covering number) bounds while these approaches are for generic SGD/SGLD and are mostly agnostic to the structure of the hypothesis class. Investigating the connections between our analysis and information-theoretic techniques is a direction for future research.

## 3 Notations and background

**Notation.** $\mathcal{X} \subseteq \mathbb{R}^d$ and $\mathcal{Y} \subseteq \mathbb{R}^p$ denote two (domain) sets. For $x \in \mathcal{X}$, let $\|x\|_1, \|x\|_2$, and $\|x\|_\infty$ denote the $\ell_1, \ell_2$, and $\ell_\infty$ norm of the vector $x$, respectively. We denote the cardinality of a set $S$ by $|S|$. The set of natural numbers smaller or equal to $m$ are denoted by $[m]$. A hypothesis is a Borel function $f : \mathbb{R}^d \to \mathbb{R}^p$, and a hypothesis class $\mathcal{F}$ is a set of hypotheses.

We also define the random counterparts of the above definitions and use an overline to distinguish them from the non-random versions. $\overline{\mathcal{X}}$ denotes the set of all random variables defined over $\mathcal{X}$ that

admit a generalized density function.[3] We sometimes abuse the notation and write $\overline{x} \in \mathcal{X}$ rather than $\overline{x} \in \overline{\mathcal{X}}$ (e.g., $\overline{x} \in \mathbb{R}^d$ is a random variable taking values in $\mathbb{R}^d$). By $\overline{y} = f(\overline{x})$ we denote a random variable that is the result of mapping $\overline{x}$ using a Borel function $f : \mathbb{R}^d \to \mathbb{R}^p$. We use $\overline{f} : \mathbb{R}^d \to \mathbb{R}^p$ to indicate that the mapping itself can be random. We use $\overline{\mathcal{F}}$ to signal that the class can include random hypotheses. We conflate the notation for random hypotheses so that they can be applied to both random and non-random inputs (e.g., $\overline{f}(\overline{x})$ and $\overline{f}(x)$).[4]

**Definition 1** (Composition of two hypothesis classes). *We denote by $h \circ f$ the function $h(f(x))$ (assuming the range of $f$ and the domain of $h$ are compatible). The composition of two hypothesis classes $\mathcal{F}$ and $\mathcal{H}$ is defined by $\mathcal{H} \circ \mathcal{F} = \{h \circ f \mid h \in \mathcal{H}, f \in \mathcal{F}\}$. Composition of classes of random hypotheses is defined similarly by $\overline{\mathcal{H}} \circ \overline{\mathcal{F}} = \{\overline{h} \circ \overline{f} \mid \overline{h} \in \overline{\mathcal{H}}, \overline{f} \in \overline{\mathcal{F}}\}$.*

The following singleton class $\overline{\mathcal{G}_\sigma}$ will be used to create noisy functions (e.g., using $\overline{\mathcal{G}_\sigma} \circ \mathcal{F}$).

**Definition 2** (The Gaussian Noise Class). *The $d$-dimensional noise class with scale $\sigma$ is denoted by $\overline{\mathcal{G}_{\sigma,d}} = \{\overline{g_{\sigma,d}}\}$. Here, $\overline{g_{\sigma,d}} : \mathbb{R}^d \to \mathbb{R}^d$ is a random function defined by $\overline{g_{\sigma,d}}(\overline{x}) = \overline{x} + \overline{z}$, where $\overline{z} \sim \mathcal{N}(\mathbf{0}, \sigma^2 I_d)$. When it is clear from the context we drop $d$ and write $\overline{\mathcal{G}_\sigma} = \{\overline{g_\sigma}\}$.*

In the rest of this section, we define the standard notion of uniform covering numbers for hypothesis classes. Intuitively, classes with larger uniform covering numbers have more capacity/flexibility, and therefore require more samples to be learned.

**Definition 3** (Covering number). *Let $(\mathcal{X}, \rho)$ be a metric space. We say that a set $A \subset \mathcal{X}$ is $\epsilon$-covered by a set $C \subseteq A$ with respect to $\rho$, if for all $a \in A$ there exists $c \in C$ such that $\rho(a, c) \leq \epsilon$. The cardinality of the smallest set $C$ that $\epsilon$-covers $A$ is denoted by $N(\epsilon, A, \rho)$ and it is referred to as the $\epsilon$-covering number of $A$ with respect to metric $\rho$.*

**Definition 4** (Extended metrics). *Let $(\mathcal{X}, \rho)$ be a metric space. Let $u = (a_1, \ldots, a_m), v = (b_1, \ldots, b_m) \in \mathcal{X}^m$ for $m \in \mathbb{N}$. The $\infty$-extended and $\ell_2$-extended metrics over $\mathcal{X}^m$ are defined by $\rho^{\infty,m}(u, v) = \sup_{1 \leq i \leq m} \rho(a_i, b_i)$ and $\rho^{\ell_2,m}(u, v) = \sqrt{\frac{1}{m} \sum_{i=1}^{m} (\rho(a_i, b_i))^2}$, respectively. We drop $m$ and use $\rho^\infty$ or $\rho^{\ell_2}$ if it is clear from the context.*

**Remark 5.** *The extended metrics are used in Definition 6 and capture the distance of two hypotheses on an input sample of size $m$. A typical example of $\rho$ is the Euclidean distance over $\mathbb{R}^p$, for which the extended metrics are denoted by $\|.\|_2^{\infty,m}$ and $\|.\|_2^{\ell_2,m}$. Unlike $\infty$-extended metric, the $\ell_2$-extended metric is normalized by $1/\sqrt{m}$, and therefore we have $\rho^{\ell_2,m}(u, v) \leq \rho^{\infty,m}(u, v)$ for all $u, v \in \mathcal{X}^m$.*

**Definition 6** (Uniform covering number). *Let $(\mathcal{Y}, \rho)$ be a metric space and $\mathcal{F}$ a hypothesis class of functions from $\mathcal{X}$ to $\mathcal{Y}$. For a set of inputs $S = \{x_1, x_2, \ldots, x_m\} \subseteq \mathcal{X}$, we define the restriction of $\mathcal{F}$ to $S$ as $\mathcal{F}_{|S} = \{(f(x_1), f(x_2), \ldots, f(x_m)) : f \in \mathcal{F}\} \subseteq \mathcal{Y}^m$. The uniform $\epsilon$-covering numbers of hypothesis class $\mathcal{F}$ with respect to metrics $\rho^\infty, \rho^{\ell_2}$ are denoted by $N_U(\epsilon, \mathcal{F}, m, \rho^\infty)$ and $N_U(\epsilon, \mathcal{F}, m, \rho^{\ell_2})$ and are the maximum values of $N(\epsilon, \mathcal{F}_{|S}, \rho^{\infty,m})$ and $N(\epsilon, \mathcal{F}_{|S}, \rho^{\ell_2,m})$ over all $S \subseteq \mathcal{X}$ with $|S| = m$, respectively.*

It is well-known that the Rademacher complexity and therefore the generalization gap of a class can be bounded based on logarithm of the uniform covering number. For sake of brevity, we defer those results to Appendix F. Therefore, our main object of interest is bounding (logarithm of) the uniform covering number. The following propositions show that there is a stark difference between classes of functions with finite range versus continuous valued functions when it comes to bounding the uniform covering number of composite classes; the proofs can be found in Appendix B.

**Proposition 7.** *Let $\mathcal{Y}$ be a finite domain ($|\mathcal{Y}| = k$) and $\rho(y, \hat{y}) = 1\{y \neq \hat{y}\}$ be a metric over $\mathcal{Y}$. For any class $\mathcal{F}$ of functions from $\mathcal{X}$ to $\mathcal{Y}$ and any class $\mathcal{H}$ of functions from $\mathcal{Y}$ to $\mathbb{R}^d$ we have $N_U(\epsilon, \mathcal{H} \circ \mathcal{F}, m, \|.\|_2^\infty) \leq N_1 . N_U(\epsilon, \mathcal{H}, m N_1, \|.\|_2^\infty)$ where $N_1 = N_U(0.5, \mathcal{F}, m, \rho^\infty)$.*

**Proposition 8.** *Let $\mathcal{F} = \{f_w(x) = wx \mid w \in (0, 1), x \in (0, 1)\}$ be a class of functions and $\mathcal{H} = \{h(y) = 1/y \mid y \in (0, 1)\}$ be a singleton class. Then, $N_U(\epsilon, \mathcal{F}, m, \|.\|_2^{\ell_2}) \leq \lceil 2/\epsilon^2 \rceil$ and $N_U(\epsilon, \mathcal{H}, m, \|.\|_2^{\ell_2}) = 1$, but $N_U(\epsilon, \mathcal{H} \circ \mathcal{F}, m, \|.\|_2^{\ell_2})$ is unbounded.*

**Proposition 9.** *For every $\epsilon > \epsilon' > 0$, there exist hypothesis classes $\mathcal{F}$ and $\mathcal{H}$ such that for every $m$ we have $N_U(\epsilon', \mathcal{H}, m, \|.\|_2^\infty) \leq m + 1$ and $N_U(\epsilon', \mathcal{F}, m, \|.\|_2^\infty) = 1$, yet $N_U(\epsilon, \mathcal{H} \circ \mathcal{F}, m, \|.\|_2^\infty) \geq 2^m$.*

---

[3]Both discrete (by using Dirac delta function) and absolutely continuous random variables admit generalized density function.

[4]Technically, we consider $\overline{f}(x)$ to be $\overline{f}(\overline{\delta_x})$, where $\overline{\delta_x}$ is a random variable with Dirac delta measure on $x$.

# 4 Covering random hypotheses

We want to establish the benefits of adding (a little bit of) noise when composing hypothesis classes. Therefore, we need to analyze classes of *random* hypotheses. One way to do this is to replace each hypothesis with its expectation, creating a deterministic version of the hypothesis class. Unfortunately, this approach misses the whole point of having noisy hypotheses (and their benefits in composition). Instead, we extend the definition of uniform covering numbers to classes of random hypotheses $\overline{\mathcal{F}}$. The following is basically the random counterpart of Definition 6.

**Definition 10** (Uniform covering number for classes of random hypotheses). *Let $(\overline{\mathcal{Y}}, \rho)$ be a metric space and $\overline{\mathcal{F}}$ a class of random hypotheses from $\overline{\mathcal{X}}$ to $\overline{\mathcal{Y}}$. For a set of random variables $\overline{S} = \{\overline{x_1}, \overline{x_2}, \ldots, \overline{x_m}\} \subseteq \overline{\mathcal{X}}$, we define the restriction of $\overline{\mathcal{F}}$ to $\overline{S}$ as $\overline{\mathcal{F}}_{|\overline{S}} = \{(\overline{f}(\overline{x_1}), \overline{f}(\overline{x_2}), \ldots, \overline{f}(\overline{x_m})) : \overline{f} \in \overline{\mathcal{F}}\} \subseteq \overline{\mathcal{Y}}^m$. Let $\Gamma \subseteq \overline{\mathcal{X}}$. The uniform $\epsilon$-covering numbers of $\overline{\mathcal{F}}$ with respect to $\Gamma$ and metrics $\rho^\infty$ and $\rho^{\ell_2}$ are defined by*

$$N_U(\epsilon, \overline{\mathcal{F}}, m, \rho^\infty, \Gamma) = \sup_{S \subseteq \Gamma, |S|=m} N(\epsilon, \overline{\mathcal{F}}_{|\overline{S}}, \rho^{\infty,m}),$$

$$N_U(\epsilon, \overline{\mathcal{F}}, m, \rho^{\ell_2}, \Gamma) = \sup_{S \subseteq \Gamma, |S|=m} N(\epsilon, \overline{\mathcal{F}}_{|\overline{S}}, \rho^{\ell_2,m}).$$

**Remark 11.** *Unlike in Definition 6 where $\rho$ is usually the $\|.\|_2$ metric in the Euclidean space, here in Definition 10 $\rho$ is defined over random variables. More specifically, we will use the Total Variation and Wasserstein metrics as concrete choices for $\rho$.*

**Remark 12.** *The specific choices that we use for $\Gamma$ are*

- $\Gamma = \overline{\mathcal{X}_d}$: *the set of all random variables defined over $\mathbb{R}^d$ that admit a generalized density function.*

- $\Gamma = \overline{\mathcal{X}_{B,d}}$: *the set of all random variables defined over $[-B, B]^d$ that admit a generalized density function.*

- $\Gamma = \overline{\Delta_d} = \{\overline{\delta_x} \mid x \in \mathbb{R}^d\}$ *and* $\Gamma = \overline{\Delta_{B,d}} = \{\overline{\delta_x} \mid x \in [-B, B]^d\}$, *where $\overline{\delta_x}$ is the random variable associated with Dirac delta measure on $x$.*

- $\Gamma = \overline{\mathcal{G}_{\sigma,d}} \circ \overline{\mathcal{X}_{B,d}} = \{\overline{g_{\sigma,d}}(\overline{x}) \mid \overline{x} \in \overline{\mathcal{X}_{B,d}}\}$: *all members of $\overline{\mathcal{X}_{B,d}}$ after being "smoothed" by adding (convolving with) Gaussian noise.*

**Remark 13.** *Some hypothesis classes that we work with have "global" covers, in the sense that the uniform covering number does not depend on $m$. We therefore use the following notation*

$$N_U(\epsilon, \overline{\mathcal{F}}, \infty, \rho^\infty, \Gamma) = \lim_{m \to \infty} N_U(\epsilon, \overline{\mathcal{F}}, m, \rho^\infty, \Gamma).$$

We now define Total Variation (TV) and Wasserstein metrics over probability measures rather than random variables, but with a slight abuse of notation we will use them for random variables too.

**Definition 14** (Total Variation Distance). *Let $\mu$ and $\nu$ denote two probability measures over $\mathcal{X}$ and let $\Omega$ be the Borel sigma-algebra over $\mathcal{X}$. The TV distance between $\mu$ and $\nu$ is defined by*

$$d_{TV}(\mu, \nu) = \sup_{B \in \Omega} |\mu(B) - \nu(B)|.$$

*Furthermore, if $\mu$ and $\nu$ have densities $f$ and $g$ then*

$$d_{TV}(\mu, \nu) = \sup_{B \in \Omega} \left| \int_B (f(x) - g(x))dx \right| = \frac{1}{2} \int_{\mathcal{X}} |f(x) - g(x)| \, dx = \frac{1}{2} \|f - g\|_1.$$

**Definition 15** (Wasserstein Distance). *Let $\mu$ and $\nu$ denote two probability measures over $\mathcal{X}$, and $\Pi(\mu, \nu)$ be the set of all their couplings. The Wasserstein distance between $\mu$ and $\nu$ is defined by*

$$d_{\mathcal{W}}(\mu, \nu) = \left( \inf_{\pi \in \Pi(\mu, \nu)} \int_{\mathcal{X} \times \mathcal{X}} \|x - y\|_2 d\pi(x, y) \right).$$

The following proposition makes it explicit that the conventional uniform covering number with respect to $\|.\|_2$ (Definition 6) can be regarded as a special case of Definition 10.

**Proposition 16.** *Let $\mathcal{F}$ be a class of (deterministic) hypotheses from $\mathbb{R}^d$ to $\mathbb{R}^p$. Then*

$$N_U(\epsilon, \mathcal{F}, m, \|.\|_2^\infty) = N_U(\epsilon, \mathcal{F}, d_\mathcal{W}^\infty, m, \overline{\Delta_d}) \text{ and } N_U(\epsilon, \mathcal{F}, m, \|.\|_2^{\ell_2}) = N_U(\epsilon, \mathcal{F}, d_\mathcal{W}^{\ell_2}, m, \overline{\Delta_d}).$$

The proposition is the direct consequence of the Definitions 6 and 10 once we note that the Wasserstein distance between Dirac random variables is just their $\ell_2$ distance, i.e., $d_\mathcal{W}(\overline{\delta_x}, \overline{\delta_y}) = \|x - y\|_2$.

# 5 Bounding the uniform covering number

This section provides tools that can be used in a general recipe for bounding the uniform covering number. The ultimate goal is to bound the (conventional) $\|.\|_2^\infty$ and $\|.\|_2^{\ell_2}$ uniform covering numbers for (noisy) compositions of hypothesis classes. In order to achieve this, we will show how one can turn TV covers into $\|.\|_2$ covers (Theorem 17) and vice versa (Corollary 21). But what is the point of going back and forth between $\|.\|_2$ and TV covers? Basically, the data processing inequality ensures an effective composition (Lemma 18) for TV covers. Our analysis goes through a number of steps, connecting covering numbers with respect to $\|.\|_2$, Wasserstein, and TV distances. The missing proofs of this section can be found in Appendix C.

The following theorem considers the deterministic class $\mathcal{H}$ associated with expectations of random hypotheses from $\overline{\mathcal{F}}$, and shows that bounding the uniform covering number of $\overline{\mathcal{F}}$ with respect to TV distance is enough for bounding the uniform covering number of $\mathcal{H}$ with respect to $\|.\|_2$ distance.

**Theorem 17** (From a TV cover to a $\|.\|_2$ cover). *Consider any class $\overline{\mathcal{F}}$ of random hypotheses $\overline{f} : \mathbb{R}^d \to [-B, B]^p$ with bounded output. Define the (nonrandom) hypothesis class $\mathcal{H} = \{h : \mathbb{R}^d \to [-B, B]^p \mid h(x) = \mathbb{E}_{\overline{f}}\left[\,\overline{f}(x)\,\right], \overline{f} \in \overline{\mathcal{F}}\}$. Then for every $\epsilon > 0$, $m \in \mathbb{N}$ these two inequalities hold:*

$$N_U(2B\epsilon\sqrt{p}, \mathcal{H}, m, \|.\|_2^\infty) \leq N_U(\epsilon, \overline{\mathcal{F}}, m, d_{TV}^\infty, \overline{\Delta_d}) \leq N_U(\epsilon, \overline{\mathcal{F}}, m, d_{TV}^\infty, \overline{\mathcal{X}_d}),$$

$$N_U(2B\epsilon\sqrt{p}, \mathcal{H}, m, \|.\|_2^{\ell_2}) \leq N_U(\epsilon, \overline{\mathcal{F}}, m, d_{TV}^{\ell_2}, \overline{\Delta_d}) \leq N_U(\epsilon, \overline{\mathcal{F}}, m, d_{TV}^{\ell_2}, \overline{\mathcal{X}_d}).$$

But what is the point of working with the TV distance? An important ingredient of our analysis is the use of data processing inequality which holds for the TV distance (see Lemma 31). The following lemma uses this fact, and shows how one can compose classes with bounded TV covers.

**Lemma 18** (Composing classes with bounded TV covers). *Let $\overline{\mathcal{F}}$ be a class of random hypotheses from $\mathbb{R}^d$ to $\mathbb{R}^p$, and $\overline{\mathcal{H}}$ be a class of random hypotheses from $\mathbb{R}^p$ to $\mathbb{R}^q$. For every $\epsilon, \epsilon' > 0$, and every $m \in \mathbb{N}$ these three inequalities hold:*

$$N_U\left(\epsilon + \epsilon', \overline{\mathcal{H}} \circ \overline{\mathcal{F}}, m, d_{TV}^\infty, \overline{\mathcal{X}_d}\right) \leq N_U\left(\epsilon', \overline{\mathcal{H}}, mN_1, d_{TV}^\infty, \overline{\mathcal{X}_p}\right).N_1,$$

$$N_U\left(\epsilon + \epsilon', \overline{\mathcal{H}} \circ \overline{\mathcal{F}}, m, d_{TV}^\infty, \overline{\Delta_d}\right) \leq N_U\left(\epsilon', \overline{\mathcal{H}}, mN_2, d_{TV}^\infty, \overline{\mathcal{X}_p}\right).N_2,$$

$$N_U\left(\epsilon + \epsilon', \overline{\mathcal{H}} \circ \overline{\mathcal{F}}, m, d_{TV}^{\ell_2}, \overline{\Delta_d}\right) \leq N_U\left(\epsilon', \overline{\mathcal{H}}, mN_3, d_{TV}^\infty, \overline{\mathcal{X}_p}\right).N_3,$$

*where $N_1 = N_U\left(\epsilon, \overline{\mathcal{F}}, m, d_{TV}^\infty, \overline{\mathcal{X}_d}\right)$, $N_2 = N_U\left(\epsilon, \overline{\mathcal{F}}, m, d_{TV}^\infty, \overline{\Delta_d}\right)$ and $N_3 = N_U\left(\epsilon, \overline{\mathcal{F}}, m, d_{TV}^{\ell_2}, \overline{\Delta_d}\right)$.*

**Remark 19.** *In Lemma 18, for $\overline{\mathcal{H}}$, we required the stronger notion of cover with respect to $\overline{\mathcal{X}_d}$ (i.e., the input to the hypotheses can be any random variable with a density function), whereas for $\overline{\mathcal{F}}$ a cover with respect to $\overline{\Delta_d}$ sufficed in some cases. As we will see below, finding a cover with respect to $\overline{\Delta_d}$ is easier since one can reuse conventional $\|.\|_2$ covers. However, finding covers with respect to $\overline{\mathcal{X}_d}$ is more challenging. In the next section we show how to do this for a class of neural networks.*

The next step is bounding the uniform covering number with respect to the TV distance (TV covering number for short). It will be useful to be able to bound TV covering number with Wasserstein covering number. However, this is generally impossible since closeness in Wasserstein distance does not imply closeness in TV distance. Yet, the following theorem establishes that one can bound the TV covering number as long as some Gaussian noise is added to the output of the hypotheses.

**Theorem 20** (From a Wasserstein cover to a TV cover). *Let $\overline{\mathcal{F}}$ be a class of random hypotheses from $\mathbb{R}^d$ to $\mathbb{R}^p$, and $\overline{\mathcal{G}_{\sigma,p}}$ be a Gaussian noise class. Then for every $\epsilon > 0$ and $m \in \mathbb{N}$ we have*

$$N_U\left(\frac{\epsilon}{2\sigma}, \overline{\mathcal{G}_{\sigma,p}} \circ \overline{\mathcal{F}}, m, d_{TV}^\infty, \overline{\mathcal{X}_d}\right) \leq N_U(\epsilon, \overline{\mathcal{F}}, m, d_\mathcal{W}^\infty, \overline{\mathcal{X}_d}),$$

$$N_U\left(\frac{\epsilon}{2\sigma}, \overline{\mathcal{G}_{\sigma,p}} \circ \overline{\mathcal{F}}, m, d_{TV}^\infty, \overline{\Delta_d}\right) \leq N_U(\epsilon, \overline{\mathcal{F}}, m, d_\mathcal{W}^\infty, \overline{\Delta_d}).$$

Intuitively, the Gaussian noise smooths out densities of random variables that are associated with applying transformation in $\overline{\mathcal{F}}$ to random variables in $\overline{\mathcal{X}_d}$ or $\overline{\Delta_d}$. As a result, the proof of Theorem 20 has a step on relating the Wasserstein distance between two smoothed (by adding random Gaussian noise) densities to their total variation distance (see Lemma 35 in Appendix C). Finally, we can use Proposition 16 to relate the Wasserstein covering number with the $\|.\|_2$ covering number. The following corollary is the result of Proposition 16 and Theorem 20 that is stated for both $d_{TV}^{\ell_2}$ and $d_{TV}^{\infty}$ extended metrics.

**Corollary 21** (From a $\|.\|_2$ cover to a TV cover). *Let $\mathcal{F}$ be a class of hypotheses $f : \mathbb{R}^d \to \mathbb{R}^p$ and $\overline{\mathcal{G}_{\sigma,p}}$ be a Gaussian noise class. Then for every $\epsilon > 0$ and $m \in \mathbb{N}$ we have*

$$N_U(\frac{\epsilon}{2\sigma}, \overline{\mathcal{G}_{\sigma,p}} \circ \mathcal{F}, m, d_{TV}^{\infty}, \overline{\Delta_d}) \leq N_U(\epsilon, \mathcal{F}, m, \|.\|_2^{\infty}),$$

$$N_U(\frac{\epsilon}{2\sigma}, \overline{\mathcal{G}_{\sigma,p}} \circ \mathcal{F}, m, d_{TV}^{\ell_2}, \overline{\Delta_d}) \leq N_U(\epsilon, \mathcal{F}, m, \|.\|_2^{\ell_2}).$$

The following theorem shows that we can get a stronger notion of TV cover with respect to $\overline{\mathcal{X}_{B,d}}$ from a $\|.\|_2$ global cover, given that some Gaussian noise is added to the output of hypotheses.

**Theorem 22** (From a global $\|.\|_2$ cover to a global TV cover). *Let $\mathcal{F}$ be a class of hypotheses $f : \mathbb{R}^d \to \mathbb{R}^p$ and $\overline{\mathcal{G}_{\sigma,p}}$ be a Gaussian noise class. Then for every $\epsilon > 0$ and $m \in \mathbb{N}$ we have*

$$N_U\left(\frac{\epsilon}{2\sigma}, \overline{\mathcal{G}_{\sigma,p}} \circ \mathcal{F}, \infty, d_{TV}^{\infty}, \overline{\mathcal{X}_{B,d}}\right) \leq N_U(\epsilon, \mathcal{F}, \infty, \|.\|_2^{\infty}).$$

The proof involves finding a Wasserstein covering number and using Theorem 20 to obtain TV covering number.

# 6 Uniform TV covers for single-layer neural networks

In this section, we study the uniform covering number of single-layer neural networks with respect to the total variation distance. This will set the stage for the next section, where we want to use the tools from Section 5 to bound covering numbers of deeper networks. We start with the following definition for the class of single-layer neural networks.

**Definition 23** (Single-Layer Sigmoid Neural Networks). *Let $\Phi : \mathbb{R}^p \to [0,1]^p$ be the element-wise sigmoid activation function defined by $\Phi((x^{(1)}, \ldots, x^{(p)})) = (\phi(x^{(1)}), \ldots, \phi(x^{(p)}))$, where $\phi(x) = \frac{1}{1+e^{-x}}$ is the sigmoid function. The class of single-layer neural networks with $d$ inputs and $p$ outputs is defined by $NET[d,p] = \{f_W : \mathbb{R}^d \to [0,1]^p \mid f_W(x) = \Phi(W^\top x), W \in \mathbb{R}^{d \times p}\}$.*

**Remark 24.** *We choose sigmoid function for simplicity, but our analysis for finding uniform covering numbers of neural networks (Theorem 25) is not specific to the sigmoid activation function. We present a stronger version of Theorem 25 in Appendix D which works for any activation function that is Lipschitz, monotone, and bounded.*

As mentioned in Remark 19, Lemma 18 requires stronger notion of covering numbers with respect to $\overline{\mathcal{X}_d}$ and TV distance. In fact, the size of this kind of cover is infinite for deterministic neural networks defined above. In contrast, Theorem 25 shows that one can bound this covering number as long as some Gaussian noise is added to the input and output of the network. The proof is quite technical, starting with estimating the smoothed input distribution ($\overline{g_\sigma}(x)$) with mixtures of Gaussians using kernel density estimation (see Lemma 58 in Appendix J). Then a cover for mixtures of Gaussians with respect to Wasserstein distance is found. Finally, Theorem 20 helps to find the cover with respect to total variation distance. For a complete proof of theorem see Appendix D.

**Theorem 25** (A global total variation cover for noisy neural networks with unbounded weights). *For every $p, d \in \mathbb{N}, \epsilon > 0, \sigma < 5d/\epsilon$ we have*

$$N_U(\epsilon, \overline{\mathcal{G}_\sigma} \circ NET[d,p], \infty, d_{TV}^{\infty}, \overline{\mathcal{G}_\sigma} \circ \overline{\mathcal{X}_{1,d}}) \leq \left(30 \frac{d^{5/2}\sqrt{\ln((5d - \epsilon\sigma)/(\epsilon\sigma))}}{\epsilon^{3/2}\sigma^2} \ln\left(\frac{5d}{\epsilon\sigma}\right)\right)^{p(d+1)}.$$

Note that the dependence of the bound on $1/\sigma$ is polynomial. he assumption $\sigma \ll 5d/\epsilon$ holds for any reasonable application (we will use $\sigma \ll 1$ in the experiments). In contrast to the analyses that exploit Lipschitz continuity, the above theorem does not require any assumptions on the norms of weights. Theorem 25 is a key tool in analyzing the uniform covering number of deeper networks.

# 7 Uniform covering numbers for deeper networks

In the following, we discuss how one can use Theorem 25 and techniques provided in Section 5 to obtain bounds on covering number for deeper networks. For a $T$-layer neural network, it is useful to separate the first layer from the rest of the network. The following theorem offers a bound on the uniform covering number of (the expectation of) a noisy network based on the usual $\|.\|_2^{\ell_2}$ covering number of the first layer and the TV covering number of the subsequent layers.

**Theorem 26.** *Let NET[$d, p_1$], NET[$p_1, p_2$], ..., NET[$p_{T-1}, p_T$] be $T$ classes of neural networks. Denote the $T$-layer noisy network by*

$$\overline{\mathcal{F}} = \overline{\mathcal{G}_\sigma} \circ NET[p_{T-1}, p_T] \circ \ldots \circ \overline{\mathcal{G}_\sigma} \circ NET[p_1, p_2] \circ \overline{\mathcal{G}_\sigma} \circ NET[d, p_1],$$

*and let $\mathcal{H} = \{h : \mathbb{R}^d \to [0,1]^{p_T} \mid h(x) = \mathbb{E}_{\overline{f}}\left[\overline{f}(x)\right], \overline{f} \in \overline{\mathcal{F}}\}$. Denote the uniform covering numbers of compositions of neural network classes with the Gaussian noise class (with respect to $d_{TV}^\infty$) as*

$$N_i = N_U\left(\frac{\epsilon}{T\sqrt{p_T}}, \overline{\mathcal{G}_\sigma} \circ NET[p_{i-1}, p_i], \infty, d_{TV}^\infty, \overline{\mathcal{G}_\sigma} \circ \overline{\mathcal{X}_{1,p_{i-1}}}\right), \ 2 \le i \le T, \tag{1}$$

*and the uniform covering number of $\overline{\mathcal{G}_\sigma} \circ NET[d, p_1]$ with respect to $\|.\|_2^{\ell_2}$ as*

$$N_1 = N_U\left(\frac{2\sigma\epsilon}{T\sqrt{p_T}}, NET[d, p_1], m, \|.\|_2^{\ell_2}\right).$$

*Then we have*

$$N_U\left(\epsilon, \mathcal{H}, m, \|.\|_2^{\ell_2}\right) \le \prod_{i=1}^{T} N_i.$$

The $\|.\|_2^{\ell_2}$ covering number of the first layer (i.e., $N_1$ in above) can be bounded using standard approaches in the literature. For instance, in Appendix H we will use the bound of Lemma 14.7 in Anthony et al. (1999). Other $N_i$'s can be bounded using Theorem 25 (See Corollary 28 in Pour and Ashtiani (2022) and Theorem 50 in Appendix H). The above bound does not depend on the norm of weights and therefore we can use it for networks with large weights.

The proof of Theorem 26 involves applying Corollary 21 to turn the $\|.\|_2$ cover of first layer into a TV cover. We then find a TV cover for rest of the network by applying Lemma 18 recursively to compose all the other layers. We will compose the first layer with the rest of the network and bound the covering number by another application of Lemma 18. Finally, we turn the TV covering number (of the entire network) back into $\|.\|_2^{\ell_2}$ covering number using Theorem 17. The complete proof can be found in Appendix E.

One can generalize the above analysis in the following way: instead of separating the first layer, one can basically "break" the network from any layer, use existing $\|.\|_2$ covering number bounds for the first few layers, and Theorem 25 for the rest. See Lemma 37 in Appendix E for details.

# 8 NVAC: a metric for comparing generalization bounds

We want to provide tools to compare different approaches in finding covering numbers and their suggested generalization bounds. First, we define the notion of a generalization bound for classification. Let $\mathcal{Y} = [k]$ and $\mathcal{F}$ be a class of functions from $\mathcal{X}$ to $\mathbb{R}^k$. Let $\mathcal{A}$ be an algorithm that receives a labeled sample $S = ((x_1, y_1), \ldots, (x_m, y_m)) \in (\mathcal{X} \times \mathcal{Y})^m$ and outputs a function $\hat{h} \in \mathcal{F}$. Note that the output of this function is a real vector so it can capture margin-based classifiers too. Let $l^{0-1} : \mathbb{R}^k \times [k] \to \{0, 1\}$ be the "thresholded" 0-1 loss function defined by $l^{0-1}(u, y) = 1\{\arg\max_i u^{(i)} \neq y\}$ where $u^{(i)}$ is the $i$-th dimension of $u$.

**Definition 27** (Generalization Bound for Classification). *A (valid) generalization bound for $\mathcal{A}$ with respect to $l^{0-1}$ and another (surrogate) loss function $l$ is a function $GB : \mathcal{F} \times (\mathcal{X} \times \mathcal{Y})^m \to \mathbb{R}$ such that for every distribution $\mathcal{D}$ over $\mathcal{X} \times \mathcal{Y}$, if $S \sim \mathcal{D}^m$, then with probability at least 0.99 (over the randomness of $S$) we have*

$$\left|\frac{1}{m}\sum_{(x,y)\in S} l(\hat{h}(x), y) - \mathbb{E}_{(x,y)\sim\mathcal{D}}\left[l^{0-1}(\hat{h}(x), y)\right]\right| \le GB(\hat{h}, S).$$

For example, $GB(\hat{h}, S) = 2$ is a useless but valid generalization bound. Various generalization bounds that have been proposed in the literature are examples of a $GB$. Note that $GB$ can depend both on $S$ (for instance on $|S|$) and on $\hat{h}$ (for example, on the norm of the weights of network).

It is not straightforward to empirically compare generalization bounds since they are often vacuous for commonly used applications. Jiang et al. (2019) address this by looking at other metrics, such as the correlation of each bound with the actual generalization gap. While these metrics are informative, it is also useful to know how far off each bound is from producing a "non-vacuous" bound (Dziugaite and Roy, 2017). Therefore, we will take a more direct approach and propose the following metric.

**Definition 28** (NVAC). *Let $\hat{h}$ be a hypothesis, $S \in (\mathcal{X} \times \mathcal{Y})^m$ a sample, and $GB$ a generalization bound for algorithm $\mathcal{A}$. Let $S^n$ denote a sample of size $mn$ which includes $n$ copies of $S$. Let $n^*$ be the smallest integer such that the following holds:*

$$GB(\hat{h}, S^{n^*}) + \frac{1}{|S^{n^*}|} \sum_{(x,y) \in S^{n^*}} l(\hat{h}(x), y) \le 1.$$

*We define NVAC to be $|S^{n^*}| = mn^*$.*

Informally speaking, NVAC is an upper bound on the minimum number of samples required to obtain a non-vacuous generalization bound. Approaches that get tighter upper bounds on covering number will generally result in smaller NVACs. In Appendix G, we will show how one can calculate NVAC using the uniform covering number bounds.

## 9 Experiments

In this section, we empirically compare different approaches in bounding the covering number using the NVAC metric. We compare the following approaches in bounding covering number: Theorem 26, Norm-based (Theorem 14.17 in Anthony et al. (1999)), Lipschitzness-based (Theorem 14.5 in Anthony et al. (1999)), Pseudo-dim-based (Theorem 14.2 in Anthony et al. (1999)), and Spectral (Bartlett et al. (2017)). More details about these bounds can be found in Appendix H.

We train fully connected neural networks on MNIST dataset. We use a network with an input layer, an output layer, and three hidden layers each containing 250 hidden neurons as the baseline architecture. See Appendix I for the details of the learning settings. The left two graphs in Figure 1 depict NVACs as functions of the depth and width of the network. It can be observed that our approach achieves the smallest NVAC. The Norm-based bound is the worst and is removed from the graph (see Appendix I). Overall, bounds that are based on the norm of the weights (even the spectral norm) perform poorly compared to those that are based on the parameter count. This is an interesting observation since we have millions of parameters ($\approx 3 \times 10^9$) in some of the wide networks and one would assume approaches based on norm of weights should be able to explain generalization behaviour better. Aside from the fact that our bound does not have any dependence on the norms of weights, there are several reasons why it performs better. First, the NVAC in Spectral and Norm-based approaches have an extra polynomial dependence on $1/\epsilon$, compared to all other approaches. Moreover, these bounds depend on product of norms and group norms which can get quite large. Finally, our method works naturally for multi-output layers, while the Pseudo-dim-based, Lipschitzness-based, and Norm-based approaches work for real-valued output (and therefore one needs to bound the cover for each output separately).

The covering number bound of Theorem 26 has a polynomial dependence on $1/\sigma$. Therefore, NVAC has a mild logarithmic dependence on $1/\sigma$ (see Appendix G for details). The third graph in Figure 1 corroborates that even a negligible amount of noise ($\sigma \approx 10^{-240}$) is sufficient to get tighter bounds on NVAC compared to other approaches. Finally, the right graph in Figure 1 shows that even with a considerable amount of noise (e.g, $\sigma = 0.2$), the train and test accuracy of the model remain almost unchanged. This is perhaps expected, as the dynamics of training neural networks with gradient descent is already noisy even without adding Gaussian noise. Overall, our preliminary experiment shows that small amount of noise does not affect the performance, yet it enables us to prove tighter generalization bounds.

**Limitations and Future Work.** Our analysis is based on the assumption that the activation function is bounded. Therefore, extending the results to ReLU neural networks is not immediate, and is left for

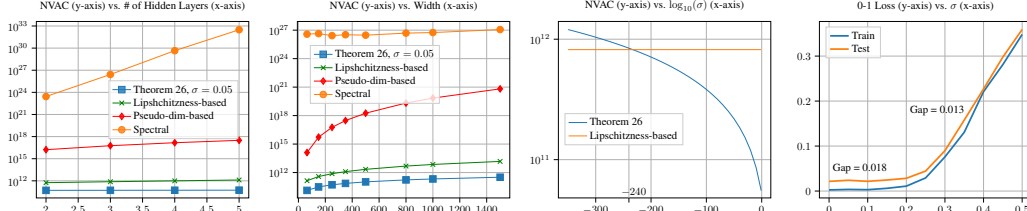

Figure 1: The left two graphs depict NVAC of different generalization bounds as a function of the number of hidden layers and width of the network. The Norm-based approach is excluded because of its excessively high NVAC (see Appendix I). The third graph plots NVAC against $\log_{10}(\sigma)$ ($\sigma$ is standard deviation of noise) for the two best approaches. The rightmost graph plots the train/test 0-1 losses for different values of $\sigma$. The gaps between the train and test losses are shown for $\sigma = 0, 0.3$.

future work. Also, our empirical analysis is preliminary and is mostly used as a sanity check. Further empirical evaluations can help to better understand the role of noise in training neural networks.

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
