# OpenReview forum: "Benefits of Additive Noise in Composing Classes with Bounded Capacity"
_NeurIPS.cc/2022/Conference — NeurIPS 2022 Accept_

### Official Review · Reviewer_xHvU · 2022-07-10

**Rating:** 6
**Confidence:** 3
**Soundness:** 3 good
**Presentation:** 3 good
**Contribution:** 3 good

**Summary:**

This paper analyzes the uniform covering number of Gaussian Noisy Neural network. It provide a generalized framework to calculate the covering number and also a metric to compare difference generalization metric.

**Questions:**

1. In the definition of extended metrics, why does L2-extended metrics require a normalization?
2. Adding noise leads to a bounded covering number, but at the same time a small capacity? Does that heart the performance of Noisy neural networks.
3. In section 7, what is the intuition of separate the first layer from others? What is the difference of analyzing the first layer and analyzing a single layer neural networks.
4. The paper can talk more on the insight from the bounded covering number. For example, can it explain why adding noise to neural nets can help reduce overfitting?
5. There are some more related literature on noisy neural networks and noisy activation function, e.g., Gulcehre et al., 2016.

**Limitations:**

1. The current analysis can only be applied on bounded activation function, such as sigmoid, softmax, tanh.

**Strengths And Weaknesses:**

Strengths:
Originality: The originality of this paper mostly comes from the new generalized mathematical framework of covering number.  Adding noise to neural nets has been studied in existing literature, but its covering number result is original.
Quality: I am not an expert in PAC learning theory but I briefly checked the proof, it looks sound to my best knowledge.
Clarity: As a math-heavy paper, this paper overall is very clear.
Significance: The generalized covering number analysis does help us understand the effect of noise in deep learning.

Weakness: My major concern is the existing framework only works for bounded activation function. This limits the applicability  of the proposed theory.

---

> ### Author Response · Authors · 2022-08-02
> **Response to Reviewer xHvU**
>
> We would like to thank the reviewer for their comments and feedback. In the following, we address the concerns and questions that are raised by the reviewer.
>
> **1.**
> > * My major concern is the existing framework only works for bounded activation function. This limits the applicability of the proposed theory.
> >* The current analysis can only be applied on bounded activation function, such as sigmoid, softmax, tanh.
>
> Making the composition tools work for ReLU functions is an exciting future research direction. It might be possible to scale the noise with the value of activation functions and use our tools to get a bound in this case. We are not aware of any general recipes for composing functions with unbounded output.
> * * *
> **2.**
> >In the definition of extended metrics, why does L2-extended metrics require a normalization?
>
>  First, we want to note that the normalized $\ell_2$ distance is used in the extended metrics to find the distance between restriction of functions to the input set while whenever we use $\lVert .\rVert_2$ we are referring to the the $\ell_2$ distance that is not normalized. For example $\lVert.\rVert_2$ is used to find the distance between $f(x_1)$ and $\hat{f}(x_1)$ while $\lVert .\rVert_2^{\ell_2}$ uses the normalized $\ell_2$ distance to find a distance between $\left(f(x_1),\ldots,f(x_m)\right)$ and $(\hat{f}(x_1),\ldots,\hat{f}(x_m))$.
>
> Using normalized $\ell_2$ distance of functions restricted to an input set is an standard approach since this is the distance that is used in classic results in chaining covering number to Rademacher complexity (see Theorems 42 and 43 in Appendix F) and providing generalization bounds. Therefore, this extended metric is the desired metric that we want to use in classic results that turn capacity (covering number) bounds into generalization bounds.
> * * *
> **3.**
> >Adding noise leads to a bounded covering number, but at the same time a small capacity? Does that heart the performance of Noisy neural networks.
>
> We showed that by adding noise we can bound the covering number of the composition, which also results in a bounded capacity as covering number is a measure of capacity. Therefore, it will improve the generalization of network and provide tighter generalization bounds. We addressed the question about performance in our experiments. We showed that in order to improve over the other bounds we need a negligible amount of noise (Third graph in Figure 1 shows that standard deviation of order $10^{-240}$ is sufficient). The negligible amount of noise is expected not to hurt the performance noticeably. Indeed, in the rightmost graph in Figure 1, we show that by adding small amounts of noise, the performance is almost the same in both training and test. Furthermore, the gap between train and test error does not increase by adding noise.
> * * *
> **4.**
> >In section 7, what is the intuition of separate the first layer from others? What is the difference of analyzing the first layer and analyzing a single layer neural networks.
>
> The reason that we separate the first layer from the other layers is twofold. First, we did not want to impose any bounds on the norms of input samples, because if we did not separate the first layer, the covering number bound would be dependent on the norm of the input. Moreover, for the covering number notions and analysis that we introduce we consider inputs of functions to be random, therefore, we separated the first layer, which has a deterministic input. Rather we wanted to add noise to the output of hidden layers and avoid making the input samples noisy. It is possible to use our analysis in the way you mentioned and avoid separating the first layer by adding noise to input samples.
> * * *
> **5.**
> >The paper can talk more on the insight from the bounded covering number. For example, can it explain why adding noise to neural nets can help reduce overfitting?
>
> As mentioned in response to your second question (in the third part of this response), by adding Gaussian noise we prove that we can bound the covering number of neural networks by composition and give capacity bounds, which are empirically better than other approaches mentioned. Being able to control the capacity by adding noise means that we are able to control and bound the generalization, which means that we may be able to reduce overfitting.
> * * *
> **6.**
> >There are some more related literature on noisy neural networks and noisy activation function, e.g., Gulcehre et al., 2016.
>
> Thanks for bringing this to our attention. There are some differences in our analysis and Gulcehre et al., 2016. More particularly, they use noise in the saturated areas of the activation functions while we always add noise to the output of functions. It would be an interesting direction to find connections between our analysis and the ideas in Gulcehre et al., 2016 to find the capacity/generalization bounds for these kinds of noise. We will refer to this work in the updated version of our paper.

---

> > ### Author Response · Authors · 2022-08-08
> > **Following up with reviewer xHvU**
> >
> > Since the author-reviewer discussion period is going to end soon we were wondering if we were able to address all your questions or if there are any further concerns that need to be discussed.

---

> > > ### Comment · Reviewer_xHvU · 2022-08-08
> > > **Thanks for the responses!**
> > >
> > > Thanks for carefully addressing all my questions. I agree extension to bounded activation function  is technically more difficult and is an important future direction. If accepted, I would like to see some empirical evaluation on ReLU to provide readers more insights in the camera ready version. I'm happy to increase my score.

---

> > > > ### Author Response · Authors · 2022-08-09
> > > > **Thank you for your comment!**
> > > >
> > > > We thank the reviewer for taking our responses into account. We are happy that you provided us with further feedbacks about ReLU activation functions and that you decided to increase the score.

---

### Official Review · Reviewer_yRWs · 2022-07-22

**Rating:** 7
**Confidence:** 3
**Soundness:** 3 good
**Presentation:** 3 good
**Contribution:** 2 fair

**Summary:**

This submission proposes a new measure of the capacity (i.e., uniform covering number) of composite classes. The motivation starts from examples where each function class has bounded capacity but the composition does not (Proposition 8,9). Due to some ill-defined capacity measures for the composition class, this paper proposes a new capacity measure defined on randomized function classes. Due to imposed randomness in the function class, TV or Wasserstein distance metrics are used to measure the distance between functions. This uniform covering number on randomized function classes has a nice property that the capacity of composite class is bounded by the covering numbers of two sub classes. Authors demonstrate the boundedness of the proposed measure for neural network function classes, and connect it to the generalization bound of deep neural networks.

**Questions:**


- As mentioned in the paper, the proposed measure cannot work with ReLU since it is not bounded. Could you clarify if there are other works that does not have an issue with ReLU activation / and if so, in which scenario does the proposed measure work better than such benchmarks?

**Limitations:**

I am happy that authors are upfront with their limitations. I do not see any negative societal impact.

**Strengths And Weaknesses:**

This paper seems to propose a new and important concept for measuring a capacity of composite class. This could be useful for giving a generalization bound for neural-network function classes with e.g., bounded activations. I have a few concerns:

- It would have been nicer if the authors could also give an explicit example of non-Lipschitz activation functions.

- Proposition 8 is a nice example for motivating the study.  But is this a right example to use? since I wondered if this example could be addressed with the newly proposed measure to show how we can benefit from it.

- Theorem 17 seems to convert the bound on a randomized class to the original function class, but requires boundedness of the output. I wonder if there are functions of interest with bounded capacity and bounded output values, but their composition does not have bounded capacity (in the example used in Proposition 8, h is unbounded).

Overall, I think the paper is well-written with clear messages. Most of research questions that can arise from the proposed concepts seem to be addressed.

---

> ### Author Response · Authors · 2022-08-02
> **Response to Reviewer yRWs**
>
> We thank the reviewer for their assessment and feedback. We address the concerns and questions raised by the reviewer in the following.
>
> **1.**
> >It would have been nicer if the authors could also give an explicit example of non-Lipschitz activation functions.
>
> We would like to note that even when the activation function has a small Lipschitz constant (such as the sigmoid activation function), the Lipschitz constant of a "layer of the network'' can be unbounded. In other words, if we look at a layer as a function, the Lipschitz constant of the layer depends on the norm of the weights/parameters. Therefore, the covering numbers obtained based on Lipschitzness have a poor dependence on the norms of the weights. In contrast, our proposed approach does not require any assumption on the magnitude of weights and works for networks with large norms. Our experiments demonstrate that the norms of the trained networks can become very large, which makes the bounds based on Lipschitzness perform worse than ours. These are reflected in Figures 1 and 2 and also in Appendix I, where the contribution of the norms of weights for a trained network is discussed.
>
> * * *
> **2.**
> > * Proposition 8 is a nice example for motivating the study. But is this a right example to use? since I wondered if this example could be addressed with the newly proposed measure to show how we can benefit from it.
> >* Theorem 17 seems to convert the bound on a randomized class to the original function class, but requires boundedness of the output. I wonder if there are functions of interest with bounded capacity and bounded output values, but their composition does not have bounded capacity (in the example used in Proposition 8, h is unbounded).
>
> The reviewer noted correctly that Proposition 8 cannot be addressed by adding noise. We have only included this proposition as a simple example of the idea that the capacity of composition can become unbounded even if the individual classes have bounded capacity.
> However, **our main message/example is conveyed in Proposition 9**.
> In the proof of Proposition 9 in Appendix B, the examples that we provide for $\mathcal{F}$ and $\mathcal{H}$ are classes of functions that have a bounded output. Therefore, Theorem 17 (and subsequent ones) can work to overcome the composition issue in proposition 9. It is worth mentioning that in Proposition 9, instead of claiming an infinite covering number, we only claim that the composition class has an exponential covering number in terms of $m$. In fact, for any class of functions with bounded outputs, the notion of covering number on $m$ input samples cannot become infinite, e.g., the class of all functions with output in $[-B,B]$ on a domain of size $m$ can be covered with $O\left(\left(2B/\epsilon\right)^m\right)$ functions. Yet, an exponential lower bound on the covering number (as in Proposition 9) implies the class of functions does not enjoy uniform convergence, which makes it challenging to provide generalization bounds. This issue can be addressed by adding noise in Proposition 9 using our techniques and get a covering number bound that grows only linearly with $m$.
>
> * * *
> **3.**
> >As mentioned in the paper, the proposed measure cannot work with ReLU since it is not bounded. Could you clarify if there are other works that does not have an issue with ReLU activation / and if so, in which scenario does the proposed measure work better than such benchmarks?
>
> As discussed in related work and appendices there are bounds that work for ReLU networks, and they depend on the number of parameter or weights of the network. With our current analysis/technique, our bound cannot be applied in this setting. We think it may be possible to extend our analysis to unbounded activation functions by scaling the noise based on the value of the activation function. We leave this for future research.
> We would like to add that we are not aware of any general recipes for composing classes of functions with potentially unbounded output (beyond the special case of neural nets).

---

### Official Review · Reviewer_Qj2x · 2022-07-22

**Rating:** 7
**Confidence:** 3
**Soundness:** 3 good
**Presentation:** 3 good
**Contribution:** 3 good

**Summary:**

The paper provides bounds on capacity of composition of noisy function classes. The paper is motivated by the fact that even when two function classes are nicely behaved and have bounded capacity, their composition might not have a bounded capacity. The paper shows that the covering number of the composition of two functions (with bounded range) is well bounded when a small amount of noise is added to the functions.

**Questions:**

Are there connections between the generalization bounds presented here and the certifiable adversarial robustness using randomized / gaussian smoothing [2]?

[2]: Cohen, J., Rosenfeld, E. and Kolter, Z., 2019, May. Certified adversarial robustness via randomized smoothing. In International Conference on Machine Learning (pp. 1310-1320). PMLR.

**Limitations:**

See the weakness section.

**Strengths And Weaknesses:**

Strengths:
1. The paper provides a novel approach to understand generalization when a small amount of noise is added while composing functions.
2. The paper provides multiple results for relating the covering numbers of noisy functions under various distance metrics like TV distance and Wasserstein distance, with those of their 'expected' non-noisy functions.
3. The idea seems intuitive to understand: adding noise will smoothen the output of the first function, thus even if it has sudden changes in its output, that gets smoothed out. This is helpful because function classes with high Lipschitz constant will also have large capacity and consequently bad generalization guarantees.
4. The paper also provides experiments that show that only a small amount of noise is needed for their bounds to be better than existing bounds.


Weaknesses:
1. The biggest problems with most existing bounds seem to be their bad (usually exponential) dependence on the depth of networks. Unfortunately, the result in this paper for deep networks (Theorem 26) also has an exponential dependence. Further, the dependence on width is also exponential here (Theorem 25). In contrast, [1] achieves a polynomial dependence in depth, although their assumptions are different.

[1]: Arora, S., Ge, R., Neyshabur, B. and Zhang, Y., 2018, July. Stronger generalization bounds for deep nets via a compression approach. In International Conference on Machine Learning (pp. 254-263). PMLR.

---

> ### Author Response · Authors · 2022-08-02
> **Response to Reviewer Qj2x**
>
> We would like to thank the reviewer for their comments and feedback. In the following, we address the concerns and questions mentioned in the review.
>
> **1.**
> >The biggest problems with most existing bounds seem to be their bad (usually exponential) dependence on the depth of networks. Unfortunately, the result in this paper for deep networks (Theorem 26) also has an exponential dependence. Further, the dependence on width is also exponential here (Theorem 25). In contrast, [1] achieves a polynomial dependence in depth, although their assumptions are different.
>
> Please note that the bound in [1] is a bound on generalization of the network, i.e., the deviation between empirical and expected error, while Theorems 25 and 26 are bounds on the covering number. Note that the generalization error can be bounded by the logarithm of covering number; see e.g., Appendix F, Theorem 43. Therefore, same as [1], our bounds in Theorems 24 and 25 result in generalization bounds that depend linearly on the depth of the network (see Theorem 48 in Appendix H). We would like to add that the bound in [1] is for the compressed network rather than the original trained network. A more discussion on this and the choice of generalization bounds that are compared in our paper can be found in Remark 45 in Appendix H.
>
> * * *
>
> **2.**
> >Are there connections between the generalization bounds presented here and the certifiable adversarial robustness using randomized / gaussian smoothing [2]?
>
> There are some differences between our analysis and that of randomized smoothing for adversarial training. Motivated by getting a more robust predictor, the randomized smoothing approach adds noise to the input of the network and chooses the class based on a majority rule. On the other hand, we add noise in between the layers of the network to bound the capacity and study uniform convergence. This being said, studying the connections between our framework and [2] is interesting and we will reference this line of work in the updated version of this paper.
>
> * * *
> [1]: Arora, S., Ge, R., Neyshabur, B. and Zhang, Y., 2018, July. Stronger generalization bounds for deep nets via a compression approach. In International Conference on Machine Learning (pp. 254-263). PMLR.
>
> [2]: Cohen, J., Rosenfeld, E. and Kolter, Z., 2019, May. Certified adversarial robustness via randomized smoothing. In International Conference on Machine Learning (pp. 1310-1320). PMLR.

---

> > ### Author Response · Authors · 2022-08-08
> > **Following Up with Reviewer Qj2x**
> >
> > We thank the reviewer again and we wonder if their concern about the dependence on depth has been addressed. We appreciate any feedback and would be happy to discuss any further questions.

---

### Meta-Review · Area_Chair_xsHj · 2022-08-24

**Recommendation:** Accept
**Confidence:** Certain

**Metareview:**

The paper present a neat idea how to use noise in order to control the capacity of composite classes

**Award:**

No

---

### Decision · Program_Chairs · 2022-09-14

Accept